# User Experiences and Attitudes Toward Sharing Wearable Activity Tracker Data with Healthcare Providers: A Cross-Sectional Study

**DOI:** 10.3390/healthcare13111215

**Published:** 2025-05-22

**Authors:** Kimberley Szeto, Carol Maher, Rachel G. Curtis, Ben Singh, Tara Cain, Darcy Beckett, Ty Ferguson

**Affiliations:** Alliance for Research in Exercise, Nutrition and Activity, UniSA Allied Health and Human Performance, University of South Australia, Adelaide, SA 5001, Australia; carol.maher@unisa.edu.au (C.M.); rachel.curtis@unisa.edu.au (R.G.C.); ben.singh@unisa.edu.au (B.S.); tara.cain@unisa.edu.au (T.C.); becdm002@mymail.unisa.edu.au (D.B.); ty.ferguson@unisa.edu.au (T.F.)

**Keywords:** wearable activity tracker, digital health, mobile health, data sharing, survey, health data, healthcare, healthcare provider

## Abstract

**Background/Objectives**: Wearable activity trackers (WATs) are increasingly used by individuals to monitor physical activity, sleep, and other health behaviors. Integrating WAT data into clinical care may offer a cost-effective strategy to support health behavior change. However, little is known about users’ willingness to share their WAT data with healthcare providers. This study aimed to explore attitudes and experiences of WAT users regarding the sharing of WAT data with healthcare providers and to examine how these vary according to user characteristics. **Methods**: An international online cross-sectional survey was conducted on adults who had used a WAT within the past three years. The survey assessed user demographics, usage patterns, experiences of sharing data with healthcare providers, and willingness or concerns regarding data sharing. Multivariate logistic regression was used to examine associations between user characteristics and data-sharing experiences or attitudes. **Results**: 447 participants completed the survey (60.0% female; 83.9% < 45 years; 60.0% from the United States). Most (94%) participants expressed willingness to share WAT data with healthcare providers, 47% had discussed it, and 43% had shared WAT data in clinical settings. Privacy was the most commonly reported concern, cited by 10% of participants. Participants with chronic health conditions were more likely to have shared or discussed WAT data, but also more likely to report concerns. Geographic differences were also observed, with Australian participants less likely to have shared or discussed their WAT data with providers, and US participants reporting fewer privacy concerns. **Conclusions**: The high willingness to share WAT data suggests that there is a possibility for integrating patient-owned WATs into clinical care. Addressing privacy concerns and equipping healthcare professionals with the skills to use WAT data will be essential to fully realize this opportunity. These findings highlight the need for further development of secure WAT systems, clinician training, and expanded evidence on WATs’ clinical utility.

## 1. Introduction

Unhealthy lifestyle behaviors such as physical inactivity and sedentary behavior [1], poor sleep quality [2], and poor diet quality [3] contribute substantially to global morbidity and mortality. Over one quarter of the global adult population does not meet recommended physical activity guidelines [1], increasing the risk of chronic diseases such as cardiovascular disease, type 2 diabetes, stroke, various cancers, and poor mental health [4]. Poor diet quality [5] and insufficient sleep [6,7] are also widespread and linked to similar health risks. The economic burden of poor lifestyle behaviors is substantial, with physical inactivity costing international healthcare systems INT 53.8 billion in 2013 [8], poor diet costing the United States approximately USD 50 billion per year [9], and the estimated overall cost of inadequate sleep costing Australia AUD 45.21 billion in 2016–2017 [10]. These figures highlight the urgent need for healthcare systems to better support healthy lifestyle behaviors.

Wearable activity trackers (WATs) have emerged as a popular, relatively low-cost tool for improving health behaviors. The international WAT market is valued at approximately USD 63 billion and projected to reach USD 352 billion by 2033 [11]. Market research estimates approximately 20% of Australian adults [12] and 39% of US adults [13] own a WAT, with most users being between 25–44 years old, tertiary educated or higher, and Apple Watches and Fitbits being the most used devices [14]. Modern day WATs have evolved from simple step counters to sophisticated devices that track numerous health metrics and integrate with smartphones to allow real time monitoring and personalized insights [15]. WATs collect real-time data on physical activity (e.g., step count, active minutes), heart rate, sleep, and oxygen saturation, supporting users in tracking and improving their behaviors [16,17], and most people report tracking and improving their health and fitness as the motivation for using WATs [14]. WATs also offer potential applications in healthcare, enabling both patients and providers to monitor and address key health behaviors [18].

Addressing lifestyle behaviors is essential in the clinical management of many chronic diseases [19]. Traditionally, healthcare providers have relied on self-reported behaviors. However, self-reporting can be inaccurate due to recall [20,21] and other systemic biases [22]. By contrast, WATs provide objective, continuous, real-world data, which can enhance healthcare decision-making [23,24]. WAT data have also been shown to support lifestyle behavior change, with evidence demonstrating effectiveness in increasing physical activity and improving clinical outcomes [16,25].

Given the widespread use of WATs, leveraging patient-owned WAT data in healthcare settings may provide a low-cost alternative to provider-issued devices. However, integrating these data requires patient willingness to share their information [26,27]. Previous research indicates that individuals with higher education, higher income, and diagnosed chronic conditions may be more open to sharing their health data [27], while underrepresented minorities and those with lower educational attainment or poorer self-rated health may be more hesitant [28]. These disparities raise concerns about potential biases in health data collection and utilization, which could exacerbate health inequities.

While research has explored general attitudes toward sharing health data, gaps remain regarding WAT-specific data sharing in healthcare contexts [26,29]. Existing studies have often focused on data sharing for research rather than clinical use [30], have been limited to single-country samples [31], or were conducted when WATs were less common [32]. Further investigation is needed to understand the extent of patient willingness to share WAT data with healthcare providers and how this varies across demographic groups.

This study aimed to examine user experiences and attitudes toward sharing WAT data with healthcare providers in an international sample. Additionally, we explored how participant characteristics (e.g., age, health status, country of residence) relate to willingness to share WAT data.

## 2. Materials and Methods

### 2.1. Study Design

This study used a cross-sectional survey design to address the aims. The data presented in this study are from a sub-set of questions in a larger cross-sectional survey that explored WAT users’ experiences related to device usability and perceived behavior change [14]. Due to the size of the larger survey and the different topic of enquiry, the subset of data presented in this study have been reported separately. Ethical approval was provided by the University of South Australia Human Research Ethics committee (Protocol number: 204908). The survey title page included study details, and completion of the survey was considered informed consent. As online surveys often collect metadata such as IP addresses and tracking links, true anonymity is difficult to guarantee. Therefore, we only collected data that were necessary for the purpose of the survey, we did not collect sensitive information, and access to study data was restricted only to the researchers.

### 2.2. Setting and Participants

An online survey was delivered via XM online survey software (Qualtrics, Provo, UT, USA, https://www.qualtrics.com/en-au/, accessed on 13 May 2025) to participants meeting the following inclusion criteria: adults 18 years and older, who either currently use or had formerly used a WAT within the past 3 years, and who used it for at least one month duration. Participants were excluded if they used health-related smartphone or computer applications without an associated WAT; used a WAT that did not measure or provide physical activity feedback; or used a WAT that did not interact with smartphone or computer applications. Data collection occurred between February and May of 2023.

Participants were recruited using two approaches, with the aim of recruiting a diverse, international sample. First, a survey link was shared on Amazon Mechanical Turk, a crowdsourcing platform. Second, the survey was shared via Facebook, including university social media pages and relevant community groups. The same survey questions were distributed to all participants regardless of which country they came from.

### 2.3. Variables

The online survey was developed based on a previous study by Maher et al. [33], along with additional items unique to the current study. To assess reliability, test-retest analysis was conducted in a pilot sample (n = 19) over a 7- to 21-day interval. Bivariate correlation coefficients for survey sections ranged from r = 0.45 to 0.79. The full survey instrument is available in Appendix A.

#### 2.3.1. Participant and Wearable Activity Tracker Characteristics

Participants’ demographic characteristics were collected, including gender, age, country of origin, self-rated health status, and the presence of chronic health conditions. Participants were asked to report if they were a current or former user of a WAT, the brand of device they used, how long and how frequently they used the WAT, and if they perceived any change in physical activity levels since using a WAT.

#### 2.3.2. Data Sharing with Healthcare Providers

Four survey items assessed participants’ experiences with WAT data sharing in healthcare settings. These items included:(1)“Have you ever used your WAT in discussions with a healthcare provider?”(2)“Have you ever directly shared health information from your WAT with a healthcare provider?”(3)“Would you be willing to share health data from a WAT with your healthcare provider?”(4)“Do you have any concerns about sharing health data from a WAT with a healthcare provider?”

Each item had a binary yes/no response option. Healthcare providers were defined as general practitioners/physicians, physiotherapists/physical therapists, exercise specialists/physiologists, or any other medical professionals providing physical activity-related recommendations or advice. Each question included an optional free-text response field for further clarification (e.g., “If yes, how?”).

### 2.4. Statistical Analysis

Participant and WAT characteristics were summarized using descriptive statistics (n and %). Responses to the four WAT data-sharing survey items were analyzed using descriptive statistics, and free-text responses were categorized thematically. Subgroup analyses were conducted using multivariate logistic regression to examine associations between WAT data-sharing attitudes and participant characteristics, including gender, age, country of residence, education level, self-rated health, and presence of chronic conditions. Given the number of comparisons, unadjusted *p*-values are reported, and results are interpreted cautiously due to the increased risk of Type I error associated with multiple testing. Events-per-variable (EPV) ratios were reviewed for each of the four models. All statistical analyses were performed in Stata 18 (StataCorp, College Station, TX, USA), with statistical significance set at *p* < 0.05. Figures were generated using MATLAB version 24.1 (MathWorks, Natick, MA, USA).

## 3. Results

### 3.1. Participants

A total of 827 survey responses were received, of which 447 were retained for analysis after removing incomplete responses (n = 14 completed <5%) and those with suspected bot activity or duplicates (n = 366). Participant characteristics are shown in Table 1. Nearly half (46.1%) were aged 25–34 years, with the majority (83.9%) under 45 years of age, and most being female (60.8%). The majority of participants resided in Australia (27.3%) or the United States (US) (60%), and 12.7% came from 31 other countries (Appendix A). Higher education qualifications were reported by 74.9% of participants, while 26.1% had up to secondary school education.

A high percentage of participants reported a self-rated health status of good or better (92.4%), with only 0.9% reporting poor health. Despite this, two-thirds (66.1%) reported having at least one chronic condition, with 38.3% reporting one condition and 28.4% reporting multiple chronic conditions.

### 3.2. Wearable Activity Tracker Characteristics

WAT-related characteristics are shown in Table 2. The majority (80.3%) were current WAT users, while 19.7% had used a WAT in the past. The most commonly used brands were Apple (44.5%), Fitbit (19.5%), and Garmin (19.9%).

Frequency of use varied, with 66.2% of participants using their WAT daily. WAT usage duration was fairly evenly distributed, with 20.6% having used their device for ≤6 months, 23.7% for 6–12 months, 23% for 1–2 years, and 19% for 3–5 years.

### 3.3. Health Data Sharing

Figure 1 shows results for the four data-sharing survey items. Of n = 413 responses, 94% (n = 388) expressed willingness to share WAT data with a healthcare provider. Of those who were not willing to provide WAT data, the most common reason cited was privacy concerns (n = 6). Of 446 responses, 47% (n = 205) had previously discussed their WAT data with a healthcare provider. The most common contexts were data measurement or tracking (n = 20), goal setting (n = 7), and discussing health concerns (n = 7). Forty-three percent (n = 187) of n = 446 respondents had directly shared data from a WAT with a healthcare provider. The most reported methods for sharing WAT data were via proprietary or third-party apps (n = 8), verbally (n = 5), or via direct messaging (n = 2). Of n = 446 responses, 26% (n = 116) had concerns about sharing data from WATs with their healthcare providers. The primary concerns included privacy (n = 10), data accuracy (n = 5), and lack of perceived benefit (n = 1).

### 3.4. Subgroup Analyses

Subgroup analyses are visually summarized in Figure 2, with full results of multivariate logistic regression provided in Appendix A. Willingness to share WAT data with healthcare providers was not significantly associated with any specific participant characteristics. Participants with chronic health conditions were more likely to have discussed (one: log-odds coefficient (coef) = 0.55 (95% CI = 0.07, 1.0), *p* = 0.026. Two or more: coef = 0.84 (95% CI = 0.29, 1.39), *p* = 0.003) or shared (one: coef = 0.59 (95% CI = 0.09, 1.09), *p* = 0.021. Two or more: coef = 0.66 (95% CI = 0.10, 1.22), *p* = 0.020) their WAT data with healthcare providers, but were also more likely to have concerns about data sharing compared to those without chronic conditions (one: coef = 0.83 (95% CI = 0.21, 1.44), *p* = 0.008. Two or more: coef = 1.09 (95% CI = 0.43, 1.75), *p* = 0.001). Australian participants were less likely to have discussed (coef = −0.83 (95% CI = −1.54, −0.12), *p* = 0.022) or shared (coef = −0.99 (95% CI = −1.72, −0.26), *p* = 0.008) their WAT data with a healthcare provider compared to participants from other countries. Australian and US participants were less likely to have concerns related to sharing their WAT data compared to participants from other countries (Australian: coef = −1.78 (95% CI = −2.62, −0.94), *p*=0.001. US: coef = −1.01 (95% CI = −1.68, −0.35), *p*=0.003). Participants aged between 35–44 were more likely to have shared their WAT data with a healthcare provider than participants aged 18–24 (coef = 0.94 (95% CI = 0.21, 1.66), *p* = 0.012), while participants aged 45–54 were more likely to have concerns about data sharing compared to younger participants (coef = 1.23 (95% CI = 0.25, 2.20), *p* = 0.013).

## 4. Discussion

This study examined attitudes and experiences of a diverse international sample of current and former WAT users on sharing their WAT health data with healthcare providers. The vast majority of participants (94%) were willing to share WAT data with healthcare providers. Despite the willingness of the sample, less than half reported experiences of either discussing (47%) or sharing (43%) WAT data, while a quarter of the sample reported having some concerns related to sharing their WAT data with healthcare providers. Concerns were mostly related to data privacy. Subgroup analyses revealed that participants with chronic health conditions were more likely to have discussed or shared their WAT data with healthcare providers yet also expressed greater concerns around privacy and data accuracy. Geographic differences also emerged, with Australian participants less likely to have shared their data and US participants less likely to report concerns.

This study reveals an opportunity for how WATs can be used in healthcare delivery to address health behavior change. Almost all respondents to the survey were willing to share their WAT data with healthcare providers. However, the majority had no prior experiences of either sharing or discussing their WAT data during clinical encounters. This finding is consistent with those of Rising et al.’s [31] survey of US adults, which also demonstrated high levels of willingness (82%) to share WAT data with healthcare providers, with less than half of their sample having used WATs or other digital health technologies during clinical encounters. Furthermore, a recent systematic review by Cascini et al. [29] found that most people were open to sharing any type of health data if it was for healthcare purposes. Together, these findings suggest WATs may be a potentially accessible source of health information that could be used to support patient care and facilitate interventions targeted at improving health behaviors. Reasons why users report limited use of WAT data during clinical encounters are currently unclear but may be due to WATs being a relatively new technological development for which its clinical utility is not yet widely recognized [34], or practitioners not yet being equipped to incorporate WAT data into healthcare delivery [35]. Nevertheless, sharing of patient-generated WAT data in clinical encounters appears to have increased over the past decade [30]. At the general population level, the popularity of WATs is substantial and continues to grow [11] and there is a large evidence base supporting their effectiveness for improving physical activity [16]. Building patient and clinician understanding of how and why to use WATs in healthcare contexts will help bridge an intention-behavior gap in sharing and using WAT data in healthcare, support their clinical uptake, and may contribute to how successfully lifestyle behaviors like physical activity and sleep are addressed in the healthcare sector.

There was a considerable proportion of participants who expressed a degree of concern about sharing their WAT data with healthcare professionals, with privacy being the most common concern that was reported. Concerns related to privacy and security when sharing any personal data with healthcare providers are common, regardless of the data type [29]. At present, many WAT devices do not meet the stringent data privacy and security requirements of healthcare [35,36]. Development of secure WAT data management solutions, such as with encryption or de-identification, may help alleviate privacy concerns and enable secure use of WAT data during clinical encounters. Additionally, appropriate data storage and management by healthcare providers is essential in ensuring privacy concerns are addressed. In general, people report greater willingness to share personal data for healthcare purposes compared to other purposes such as commercial use [29], and greater willingness where there is data governance and oversight from government or public healthcare as opposed to private businesses and insurers [28]. Ensuring that the use and governance of patient data prioritizes patient privacy and remains for the purpose of supporting healthcare delivery is critical in fostering patient willingness to share personal information and health data with providers, regardless of data type.

Our study did not reveal that any particular demographic factor influenced the likelihood of participants reporting their willingness to share WAT data with healthcare providers. This is similar to findings from a survey of US adults regarding WAT data sharing, which also found no significant relationship between willingness to share WAT data with healthcare providers and sociodemographic factors [29]. Other studies that have explored data sharing attitudes for a range of health data types have demonstrated that demographic factors such as education, age, race, income, and self-rated health status can influence willingness to share personal data with healthcare providers [26,27,28]. It is possible that, unlike with other types of health data (e.g., genomic data or personal health and medical history) [29], willingness to share WAT data is not influenced by sociodemographic factors.

Participants with chronic health conditions were more likely to have had discussions about or shared their WAT data with healthcare providers and were also more likely to have concerns about sharing their data. It is likely that respondents with chronic conditions have had more interactions with healthcare providers generally [37,38,39], giving rise to more opportunities to discuss or share their WAT data. The increased likelihood of having concerns about sharing WAT data amongst this subset of participants may also represent an availability heuristic, where participants who have had experience discussing or sharing their WAT data have considered associated personal risks more so than those who have not [40].

### 4.1. Strengths and Limitations

There are some key strengths to this study. First, we explored data sharing attitudes and experiences specifically for WAT data. While there is a considerable evidence base exploring data sharing attitudes and experiences for general health data [29], there is a dearth of studies focused on health data sharing related to WATs specifically. As the consumer WAT market continues to grow, user data collected by WATs can provide useful insights relevant to healthcare provision, and understanding data sharing attitudes and concerns for these devices can inform how this is approached. Another key strength is in the inclusion of both current and former WAT users. Including former WAT users allowed us to capture attitudes and experiences of individuals who may have abandoned using WATs due to having negative experiences. The inclusion of former WAT users may have potentially reduced self-selection bias influencing the study findings. A final strength of this study was using two different recruitment methods. Using two approaches enabled recruitment of an international sample, with diverse representation of demographics including age ranges, education levels, genders, and health statuses, and capture of a wider range of perspectives.

This study also has some limitations. As with many online surveys, a limitation of this study is the potential for self-selection bias, whereby individuals with a particular interest in the topic may have been more likely to participate. While 20% of the included sample were former WAT users, it is still possible that self-selection bias may have influenced the findings. The nature of the survey study design may have also introduced recall bias. It is possible that participants may not have accurately recalled occurrences or details of clinical encounters involving discussion or sharing of WAT data, particularly among former WAT users, and especially if such interactions were brief, infrequent, or occurred a long time prior to completing the survey. While this may have introduced some inaccuracy, we anticipate that any effects would be non-systematic and unlikely to bias the results in a particular direction. Furthermore, Mechanical Turk and Facebook recruitment might have skewed the sample toward more tech-savvy and younger users. Though we recruited a global sample, some populations were over-represented. In particular, the sample in this study were mostly younger than 45 years, tertiary educated, and came from either Australia or the United States. The demographic characteristics of this sample means that the experiences and attitudes to data sharing identified may not be generalizable for the wider population or individuals from broader age, income, or education groups. Exploring experiences and attitudes on this topic amongst individuals with different demographic characteristics or in less represented countries may provide more varied insights. Finally, it is possible that some participants started using WATs following discussions with healthcare providers rather than before, and due to the nature of the study design used, our results do not describe if participants’ WAT use was due to such discussions.

From an analysis perspective, the model for participant concerns had an events-per-variable ratio below the commonly recommended threshold of 10. While the model converged and produced interpretable results, this low EPV suggests that estimates from the model examining participant concerns may be less stable and should be interpreted cautiously.

### 4.2. Implications and Future Directions

This study indicates that patients are willing and inclined to share data from their WATs with healthcare providers where there are opportunities to do so. Using WAT devices that patients own provides an efficient and low-cost alternative to healthcare services providing patients with devices. However, realizing this potential requires addressing user and provider barriers, and investment in technology, infrastructure, data security, and clinician training. To use WAT data for clinical decision making, it is essential that data are sufficiently accurate. Currently, the validity and reliability of WAT data can vary across different devices and systems [41]. Identifying which devices meet the needs of healthcare delivery is critical to incorporate them into care, as is the development of devices and systems that provide accurate data. Evaluating the cost of integrating patient-collected WAT data into healthcare is a key step, as is addressing privacy concerns related to sharing WAT data, and obstacles related to awareness and skill will support users and providers in using patient-provided WAT data to enhance healthcare delivery. Improving data privacy and security mechanisms in WATs and their software will be critical in addressing privacy concerns as well as supporting digital transfer of data. Second, future research should examine the clinical utility of WATs for diverse clinical populations and purposes, such as with prospective cohort studies or randomized trials. Expanding the evidence base on how WATs can be used for monitoring and detecting health conditions and how using them to improve health behaviors may lead to secondary improvements in clinical outcomes will contribute to the already large evidence base demonstrating their ability to improve health behaviors like physical activity. Third, healthcare providers need greater training and support to promote physical activity as a routine part of care. The World Health Organization’s Global Action Plan on Physical Activity highlights this as a global priority, noting that many providers feel underprepared to counsel patients on physical activity [1]. Finally, it is necessary that healthcare providers are aware of how WATs can enhance healthcare delivery and are equipped with the skills to use them accordingly [35]. Disseminating and supplying healthcare providers with information on why and how to use WATs will be an important step in embracing this technology in healthcare delivery.

## 5. Conclusions

This study provides insights into the attitudes and willingness of WAT users regarding sharing their WAT data with healthcare providers. The findings show that most WAT users are willing to share their WAT data with providers, yet less than half have had experience doing so, and some users have data sharing concerns, mostly related to privacy. These findings highlight an opportunity to leverage patient-owned WATs to enhance healthcare delivery by providing an efficient and economical approach to monitoring and promoting health behaviors like physical activity and sleep.

To fully embrace this opportunity, there is a need for continued development of WAT devices and software that offer data privacy and security features aligned with healthcare system requirements. Future work should also expand the evidence base for how WATs can be used to monitor and manage a range of health conditions, explore if sharing WAT data in healthcare leads to improved clinical decision making and patient outcomes as well as providing healthcare professionals with the training, tools, and guidance necessary to incorporate WAT data effectively into routine care.

## Figures and Tables

**Figure 1 healthcare-13-01215-f001:**
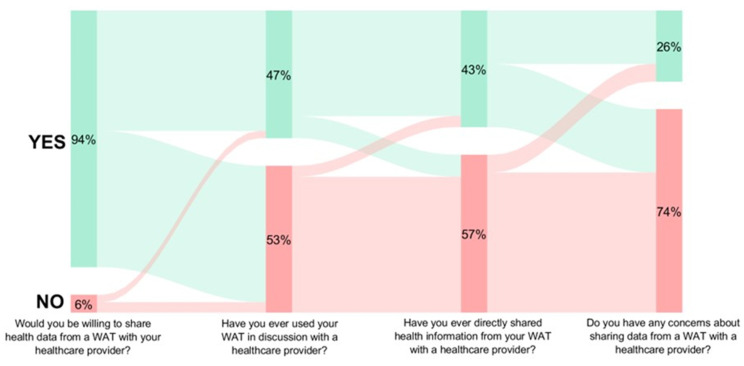
User experiences of wearable activity trackers with healthcare providers.

**Figure 2 healthcare-13-01215-f002:**
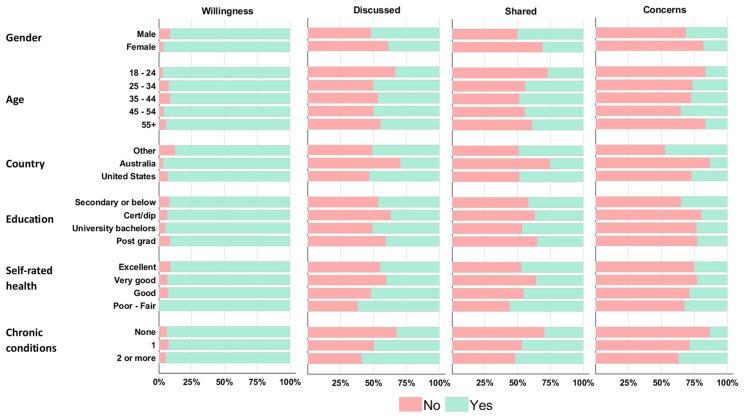
Wearable activity tracker data sharing by participant characteristics subgroup.

**Table 1 healthcare-13-01215-t001:** Participant Characteristics (n = 447).

		n	(%)
Gender	Female	268	(60.0)
	Male	178	(39.8)
	Non-binary/third gender	1	(0.2)
Age (in years)	18–24	66	(14.8)
	25–34	204	(45.6)
	35–44	105	(23.5)
	45–54	54	(12.1)
	≥55	18	(4)
Country	United States of America	268	(60.0)
	Australia	122	(27.3)
	Other	57	(12.7)
Education	Secondary school or below	117	(26.1)
	Vocational qualification (i.e., certificate or diploma)	46	(10.3)
	University bachelor’s degree	213	(47.7)
	Post graduate degree (e.g., Master’s degree, PhD)	71	(15.9)
Self-rated health	Excellent	80	(17.9)
	Very good	190	(42.5)
	Good	143	(32.0)
	Fair	30	(6.7)
	Poor	4	(0.9)
Chronic conditions	None	149	(33.3%)
	1	171	(38.3%)
	2 or more	127	(28.4%)

**Table 2 healthcare-13-01215-t002:** Wearable activity tracker characteristics.

		n	(%)
User status	Current	359	(80.3)
	Former	88	(19.7)
Wearable brand	Apple	199	(44.5)
	Fitbit	87	(19.5)
	Garmin	89	(19.9)
	Samsung	46	(10.3)
	Other	26	(5.8)
Duration of use	≤6 months	92	(20.6)
	>6–12 months	106	(23.7)
	>1–2 years	103	(23.0)
	3–5 years	85	(19.0)
	≥6 years	39	(6.5)
	Not reported	22	(7.2)
Frequency of use (% of current users, n = 359)	Every day including overnight	90	(25)
	Every day during waking hours	148	(41.2)
	Once a week	69	(19.2)
	When I remember	27	(8)
	Other	5	(1.4)
	Not reported	20	(3.6)
Perceived change in physical activity since using wearable	Increased and maintained	130	(29.1)
	Constantly increased activity	165	(36.9)
	No change	81	(18.1)
	Increased but unable to maintain	43	(9.6)
	Other	6	(1.3)
	Not reported	22	(4.9)

## Data Availability

The datasets used and/or analyzed during this study are available from the corresponding author on reasonable request.

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
