# Peer review of "User Experiences and Attitudes Toward Sharing Wearable Activity Tracker Data with Healthcare Providers: A Cross-Sectional Study"

_healthcare, 2025, doi:10.3390/healthcare13111215_

Round 1
Reviewer 1 Report
Comments and Suggestions for Authors
Comments:
- In Table 1, the Age units are not written; kindly add "Age (in years)".
- Most of the respondents (nearly 90%) are from the USA and Australia. Kindly clarify the same whether the survey questions were circulated to both countries. Discuss the same thing in the discussion section as well.
Author Response
Dear Reviewer 1,
COMMENT 1: In Table 1, the Age units are not written; kindly add "Age (in years)".
Response to reviewer:
This has been added.
Change and location in manuscript:
Table 1, page 4-5
COMMENT 2:
Most of the respondents (nearly 90%) are from the USA and Australia. Kindly clarify the same whether the survey questions were circulated to both countries. Discuss the same thing in the discussion section as well.
Response to reviewer:
Thank you for this comment. All participants were provided with the same survey questions, as the questions were broad and not focused on location-specific processes. We have added this detail in the methods section, and have addressed the possibility of exploring this topic in different locations less represented in this study in the Discussion section.
Change and location in manuscript:
“The same survey questions were distributed to all participants regardless of which country they came from.”
- Lines 115-116, page 3
“Though we recruited a global sample, some populations were over-represented. In particular, the sample in this study were mostly younger than 45 years, at least tertiary educated, and came from either Australia or the United States. The demographic characteristics of this sample means that the experiences and attitudes to data sharing identified may not be generalizable for the wider population or individuals from broader age, income or education groups. Exploring experiences and attitudes on this topic amongst individuals with different demographic characteristics or in less represented countries may provide more varied insights.”
- Lines 320-328, page 9
Reviewer 2 Report
Comments and Suggestions for Authors
Thank you for the opportunity to review your work. Please, see below few comments you might consider or clarify:
1. Study Design & Interpretation
a. Cross-Sectional Constraints
Because your study is cross-sectional, all findings represent associations at a single point in time. You should:
Avoid causal language. Rephrase statements like “integrating WAT data supports health behavior change” to “is associated with willingness to share data.”
Acknowledge temporality limitations. Clarify that you cannot know whether WAT use preceded—or resulted from—data‐sharing discussions.
Discuss recall bias. Participants reported lifetime “ever shared” experiences over a three-year window; memory errors may inflate or misdate sharing events.
b. Misinterpretation of Willingness
Ninety-four percent reported willingness to share, yet only 43–47% had actually done so. High stated willingness in a survey does not guarantee real-world behavior. Please temper claims about “untapped potential” with the well-known “intention–behavior gap.”
2. Sampling & Generalizability
a. Online Convenience Sample
Recruitment via Amazon Mechanical Turk and Facebook skews toward younger, more tech-savvy, and higher-educated individuals (e.g., 83.8% under 45 years; 74.5% with a tertiary degree)
. As such:
Limit your generalizations. Acknowledge that older adults, lower-income groups, and non-digital natives are underrepresented.
Report recruitment flow. If available, state how many people saw the invitation, how many started vs. completed the survey, and reasons for non-participation to help readers assess non-response bias.
b. Self-Selection & Recall
Former users (19.7%) may differ systematically from current users and may misremember past clinical interactions. Discuss how this self-selection and recall may bias your estimates of sharing prevalence.
3. Statistical Analysis & Reporting
a. Multiple Comparisons
You ran multivariate logistic regressions for four outcomes across six predictors (24 tests) at α=0.05 without adjustment. This inflates Type I error risk. Either apply a Bonferroni or false-discovery correction or label these analyses as exploratory and interpret p-values cautiously.
b. Omission of Effect Sizes
Key odds ratios and 95% confidence intervals reside only in Supplementary File 2. Summarize the most important adjusted ORs (e.g., chronic condition → shared data OR = X) in the main Results to improve reader accessibility.
c. Model Fit & Sparse Data
Some subgroups (e.g., ≥55 years, non-binary gender) are very small, risking unstable estimates and over-fitting. Provide events-per-variable information or goodness-of-fit statistics, and consider collapsing sparse categories or noting limited power.
d. Handling Missing Data
You do not describe how missing responses were handled. State whether you used listwise deletion, imputation, or another approach, and report the number of complete cases per analysis.
4. Tables & Data Presentation
a. Table 1 (Participant Characteristics)
The ≥55 years category is shown as 18 (5.1%), but 18/447 ≈ 4.0%
Ensure each category’s percentage is calculated as (n ÷ 447 × 100) so that totals sum to 100%.
b. Table 2 (WAT Characteristics)
Frequency-of-use percentages appear to use the current-user denominator (n = 359) but are mis-computed (e.g., 148/359 ≈ 41.2%, not 43.7%)
In your column header, specify “% of current users (n = 359)”, then recalculate so the categories sum to exactly 100%.
c. Missing vs. “Other”
You conflate “Other / not reported” for brand and “Not reported” for duration. Distinguish true missing data (“No response”) from an “Other” free-text choice, and report missing counts separately.
5. Framing & Clinical Implications
a. Temper “Low-Cost” & “Clinical Utility” Claims
While consumer-owned WATs may reduce device procurement costs, integration requires investment in IT infrastructure, data security, and clinician training.
Rephrase “WATs provide a cost-effective strategy” to “may offer a lower-cost alternative for data collection, pending evaluation of integration costs.”
b. Call for Prospective Research
Your cross-sectional snapshot cannot determine whether data-sharing leads to better outcomes. Conclude by recommending prospective cohorts or randomized trials to test whether sharing WAT data actually improves clinical decision-making and patient health.
Comments on the Author’s Self-citations:
Refs. 13, 22, 31, 32, 34, 35, 42 and 43 which is a self-citation rate of about 18.6% (8/43).
After re-examining the references, I believe the author should cite only the three systematic reviews and meta-analyses, as they comprehensively cover the themes of the other articles (13,22, 42).
Looking forward to reading your revised work,
Best wishes
Author Response
Dear Reviewer 2,
COMMENT 1: Cross-Sectional Constraints
Because your study is cross-sectional, all findings represent associations at a single point in time. You should:
- Avoid causal language. Rephrase statements like “integrating WAT data supports health behavior change” to “is associated with willingness to share data.”
Response to reviewer:
Thank you for your feedback. The parts of the paper where we have stated that WAT data can be used to support behaviour change are in the Background section of the Abstract and the Introduction, and citations for papers are included that support this (line 70, citations [16, 25]). Where it is suitable to edit the wording to use less causal language, we have done so.
Change and location in manuscript:
“This study reveals an opportunity for how WATs can be used in healthcare delivery to address health behavior change.”
Line 234-235, page 7
COMMENT 2: Acknowledge temporality limitations. Clarify that you cannot know whether WAT use preceded—or resulted from—data‐sharing discussions.
Response to reviewer:
Thank you for this suggestion. We have incorporated this into the Discussion.
Change and location in manuscript:
“Finally, it is possible that some participants started using WATs following discussions with healthcare providers rather than before, and due to the nature of the study design used, our results do not describe if participants’ WAT use was due to such discussions.”
- Line 328-331, page 9
COMMENT 3: Discuss recall bias. Participants reported lifetime “ever shared” experiences over a three-year window; memory errors may inflate or misdate sharing events.
Response to reviewer:
Thank you for this suggestion. As requested, we have expanded the discussion limitations section to acknowledge potential recall issues.
Change and location in manuscript:
“It is possible that participants may not have accurately recalled occurrences or details of clinical encounters involving discussion or sharing of WAT data, particularly among former WAT users, and especially if such interactions were brief, infrequent, or occurred a long time prior to completing the survey.”
- Line 314-317, page 9
COMMENT 4: Misinterpretation of Willingness
Ninety-four percent reported willingness to share, yet only 43–47% had actually done so. High stated willingness in a survey does not guarantee real-world behavior. Please temper claims about “untapped potential” with the well-known “intention–behavior gap.”
Response to reviewer:
We recognise that willingness does not guarantee real-world behaviour, and the wording “untapped potential” could be improved. In addition to an intention-behaviour gap, we also believe that the gap between willingness to share and actually sharing is influenced by the broader barriers described in the discussion (healthcare provider awareness and skill with WATs, suboptimal technology integration), as well as an To address the reviewer’s request and to temper claims on “untapped potential”, we have added additional detail and edited the suggested wording:
Change and location in manuscript:
“The high willingness to share WAT data suggests there is a possibility for integrating patient-owned WATs into clinical care”
- Line 29, page 1
“Together, these findings suggest WATs may be a potentially accessible source of health information that could be used to support patient care and facilitate interventions targeted at improving health behaviors.”
- Line 243-245, page 8
“Building patient and clinician understanding of how and why to use WATs in healthcare contexts will help bridge an intention-behavior gap in sharing and using WAT data in healthcare, support their clinical uptake, and may contribute to how successfully lifestyle behaviors like physical activity and sleep are addressed in the healthcare sector.”
- Lines 253-256, page 8
COMMENT 5: Online Convenience Sample
Recruitment via Amazon Mechanical Turk and Facebook skews toward younger, more tech-savvy, and higher-educated individuals (e.g., 83.8% under 45 years; 74.5% with a tertiary degree). As such:
- Limit your generalizations. Acknowledge that older adults, lower-income groups, and non-digital natives are underrepresented.
Response to reviewer:
We have provided further detail that the sample recruited using these approaches and the results may not be generalizable to the wider population.
Change and location in manuscript:
“Though we recruited a global sample, some populations were over-represented. In particular, the sample in this study were mostly younger than 45 years, tertiary educated, and came from either Australia or the United States. The demographic characteristics of this sample means that the experiences and attitudes to data sharing identified may not be generalizable for the wider population or individuals from broader age, income or education groups.”
- Line 320-326, page 9
COMMENT 6: Report recruitment flow. If available, state how many people saw the invitation, how many started vs. completed the survey, and reasons for non-participation to help readers assess non-response bias.
Response to reviewer:
The recruitment approach did not allow us to assess how many people saw the invitation to participate or assess non-response bias, as we did not contact individual participants. As this study used a subset of data collected from a larger survey, incomplete surveys are included in the analyses where data of interest was complete. This means that the amount of completed surveys is lower than the sample used, and describing incomplete surveys will be confusing to the reader. To address the reviewer’s suggestion to provide more detail on recruitment flow, we have provided further detail in the results section on the number of survey responses that we were able to use in the final data set.
Change and location in manuscript:
“A total of 827 survey responses were received, of which 447 were retained for analysis after removing incomplete responses (n=14 completed <5%) and those with suspected bot activity or duplicates (n =366). "
- Line 159-161, page 4
COMMENT 7: Self-Selection & Recall
Former users (19.7%) may differ systematically from current users and may misremember past clinical interactions.
Response to reviewer:
As requested, we have added these limitations in the discussion.
Change and location in manuscript:
“As with many online surveys, a limitation of this study is the potential for self-selection bias, whereby individuals with a particular interest in the topic may have been more likely to participate.”
- Line 309-311, page 9
“It is possible that participants may not have accurately recalled occurrences or details of clinical encounters involving discussion or sharing of WAT data, particularly among former WAT users, and especially if such interactions were brief, infrequent, or occurred a long time prior to completing the survey.”
- Line 314-317, page 9
COMMENT 8: Multiple Comparisons
You ran multivariate logistic regressions for four outcomes across six predictors (24 tests) at α=0.05 without adjustment. This inflates Type I error risk. Either apply a Bonferroni or false-discovery correction or label these analyses as exploratory and interpret p-values cautiously.
Response to reviewer:
These analyses were indeed exploratory. The following clarification has been added to the manuscript.
Change and location in manuscript:
“Given the number of comparisons, unadjusted p-values are reported, and results are interpreted cautiously due to the increased risk of Type I error associated with multiple testing.”
- Line 151-153, page 4
COMMENT 9: Omission of Effect Sizes
Key odds ratios and 95% confidence intervals reside only in Supplementary File 2. Summarize the most important adjusted ORs (e.g., chronic condition → shared data OR = X) in the main Results to improve reader accessibility.
Response to reviewer:
As suggested, log-odds coefficients and 95% confidence intervals have been added throughout the subgroup analysis paragraph
Change and location in manuscript:
“Participants with chronic health conditions were more likely to have discussed (one: log-odds coefficient (coef) = 0.55 [95% CI = 0.07, 1.0], p = 0.026. Two or more: coef = 0.84 [95% CI = 0.29, 1.39], p = 0.003) or shared (one: coef = 0.59 [95% CI = 0.09, 1.09], p = 0.021. Two or more: coef = 0.66 [95% CI = 0.10, 1.22], p = 0.020) their WAT data with healthcare providers, but were also more likely to have concerns about data sharing compared to those without chronic conditions (one: coef = 0.83 [95% CI = 0.21, 1.44], p = 0.008. Two or more: coef = 1.09 [95% CI = 0.43, 1.75], p = 0.001). Australian participants were less likely to have discussed (coef = -0.83 [95% CI = -1.54, -0.12], p = 0.022) or shared (coef = -0.99 [95% CI = -1.72, -0.26], p = 0.008) their WAT data with a healthcare provider compared to participants from other countries. Australian and US participants were less likely to have concerns related to sharing their WAT data compared to participants from other countries (Australian: coef = -1.78 [95% CI = -2.62, -0.94], p=0.001. US: coef = -1.01 [95% CI = -1.68, -0.35], p=0.003). Participants aged between 35-44 were more likely to have shared their WAT data with a healthcare provider than participants aged 18-24 (coef = 0.94 [95% CI = 0.21, 1.66], p = 0.012), while participants aged 45-54 were more likely to have concerns about data sharing compared to younger participants (coef = 1.23 [95% CI = 0.25, 2.20], p = 0.013).”
- Line 203-218, page 6-7
COMMENT 10: Model Fit & Sparse Data
Some subgroups (e.g., ≥55 years, non-binary gender) are very small, risking unstable estimates and over-fitting. Provide events-per-variable information or goodness-of-fit statistics, and consider collapsing sparse categories or noting limited power.
Response to reviewer:
Thank you for the suggestion. The non-binary gender category was not included in the final regression models due to small sample size and automatic exclusion by Stata.
Model fit statistics including pseudo-R², LR chi², and log-likelihood values are already reported in the results tables for transparency. While we did not explicitly include events-per-variable ratios, we reviewed them across models and found them to be acceptable for 3 of the 4 models, with the “concerns” model = 8.3. We have added text to the Methods and Discussion.
Change and location in manuscript:
“Events-per-variable (EPV) ratios were reviewed for each of the four models.”
- Line 153-156, page 4
“From an analysis perspective, the model for participant concerns had an events-per-variable ratio below the commonly recommended threshold of 10. While the model converged and produced interpretable results, this low EPV suggests that estimates from the model examining participant concerns may be less stable and should be interpreted cautiously.“
- Line 332-336, page 9
COMMENT 11: Handling Missing Data
You do not describe how missing responses were handled. State whether you used listwise deletion, imputation, or another approach, and report the number of complete cases per analysis
Response to reviewer:
Where there were missing responses, we did not include these participants in the analyses. To make this clearer in the manuscript, we have added in the number of responses that results are based on for each question in the results section. For subgroup analyses, the number of observations is already included in the tables in supplementary files.
Change and location in manuscript:
“Of n = 413 responses, 94% (n = 388) expressed willingness to share WAT data with a healthcare provider. Of those who were not willing to provide WAT data, the most common reason cited was privacy concerns (n = 6). Of 446 responses, 46% (n = 205) had previously discussed their WAT data with a healthcare provider. The most common contexts were data measurement or tracking (n = 20), goal setting (n = 7) and discussing health concerns (n = 7). Forty-two percent (n = 187) of n = 446 respondents had directly shared data from a WAT with a healthcare provider. The most reported methods for sharing WAT data were via proprietary or third-party apps (n = 8), verbally (n = 5) or via direct messaging (n = 2). Of n = 446 responses, 26% (n = 116) had concerns about sharing data from WATs with their healthcare providers.”
- Lines 183-193, page 6
COMMENT 12: Table 1 (Participant Characteristics)
- The ≥55 years category is shown as 18 (5.1%), but 18/447 ≈ 4.0%
- Ensure each category’s percentage is calculated as (n ÷ 447 × 100) so that totals sum to 100%.
Response to reviewer:
Thank you for highlighting this. We have adjusted the percentages in Table 1
Change and location in manuscript:
- Table 1, page 4-5
COMMENT 13: Table 2 (WAT Characteristics)
- Frequency-of-use percentages appear to use the current-user denominator (n = 359) but are mis-computed (e.g., 148/359 ≈ 41.2%, not 43.7%) - In your column header, specify “% of current users (n = 359)”, then recalculate so the categories sum to exactly 100%.
Response to reviewer:
These have been recalculated.
Change and location in manuscript:
Table 2, page 5-6
COMMENT 14: Missing vs. “Other”
You conflate “Other / not reported” for brand and “Not reported” for duration. Distinguish true missing data (“No response”) from an “Other” free-text choice, and report missing counts separately
Response to reviewer:
Thank you for this suggestion. Participants were given the option to provide which ‘Other’ brand device they used, however no responses were provided for this. For clarity we have removed ‘not reported’ from Table 2.
Change and location in manuscript:
Table 2, page 5-6
COMMENT 15: Temper “Low-Cost” & “Clinical Utility” Claims
- While consumer-owned WATs may reduce device procurement costs, integration requires investment in IT infrastructure, data security, and clinician training.
- Rephrase “WATs provide a cost-effective strategy” to “may offer a lower-cost alternative for data collection, pending evaluation of integration costs.”
Response to reviewer:
Thank you for these considerations. We have incorporated your suggestions and addressed this in the Discussion.
Change and location in manuscript:
“However, realizing this potential requires addressing user and provider barriers, and investment in technology, infrastructure, data security and clinician training. To use WAT data for clinical decision making, it is essential that data is sufficiently accurate. Currently, the validity and reliability of WAT data can vary across different devices and systems [41]. Identifying which devices meet the needs of healthcare delivery is critical to incorporate them into care, as is development of devices and systems that provide accurate data. Evaluating the cost of integrating patient-collected WAT data into healthcare against is a key step, as is addressing privacy concerns related to sharing WAT data, and obstacles related to awareness and skill will support users and providers in using patient-provided WAT data to enhance healthcare delivery.”
- Lines 341-350, page 10
COMMENT 16: Call for Prospective Research
- Your cross-sectional snapshot cannot determine whether data-sharing leads to better outcomes. Conclude by recommending prospective cohorts or randomized trials to test whether sharing WAT data actually improves clinical decision-making and patient health.
Response to reviewer:
Thank you for this suggestion. We have addressed this in the Discussion and in the Conclusion.
Change and location in manuscript:
“Secondly, future research should examine the clinical utility of WATs for diverse clinical populations and purposes such as with prospective cohort studies or randomized trials.”
- Line 352-354, page 10
“Future work should also expand the evidence base for how WATs can be used to monitor and manage a range of health conditions, explore if sharing WAT data in healthcare leads to improved clinical decision making and patient outcomes as well as provide healthcare professionals with the training, tools, and guidance necessary to incorporate WAT data effectively into routine care.”
- Lines 377-381, page 10
COMMENT 17: Refs. 13, 22, 31, 32, 34, 35, 42 and 43 which is a self-citation rate of about 18.6% (8/43).
After re-examining the references, I believe the author should cite only the three systematic reviews and meta-analyses, as they comprehensively cover the themes of the other articles (13,22, 42).
Response to reviewer:
We have removed the following citations 22, 35,42, 43 to reduce the self-citation rate.
Citation [34] was replaced with a different article (now citation number [35]).
The remaining citations were kept as they refer to the full study that this survey was conducted as part of [14] and the earlier iteration of this work [33], and included an umbrella review that was appropriate for the text being supported [16].
Change and location in manuscript:
“At present, many WAT devices do not meet the stringent data privacy and security requirements of healthcare [35, 36].”
- 261-262, page 8
“35. Smuck, M, Odonkor, CA, Wilt, JK, Schmidt, N, Swiemik, MA. The emerging role of wearables: factors for successful implementation in healthcare. MPJ digital medicine. 2021;4(1):45.”
- Line 482-483. Page 13
Reviewer 3 Report
Comments and Suggestions for Authors
Dear authors,
I appreciate the opportunity to review your manuscript. I consider your research relevant and timely.
The aim of the study is to explore attitudes and experiences of WAT users regarding the sharing of WAT data with healthcare providers,and examine how these vary according to user characteristics.
After reading and analyzing your document, I would like to send you a series of comments/suggestions.
The topic is presented in the introduction in an orderly and organized manner, which aids understanding. You have used a considerable number of bibliographic references. However, a high number of them correspond to old publications, more than 5 years old. Therefore, you should search for more recent references and update them.
Furrthermore, it might be interesting to include more information in the introduction about the number of WAT users, both globally and in the countries participating in the study, both in the general population and in patients with chronic illnesses, as well as the personal characteristics of the users of these devices.
In Results, you group the data by country of the participant: United States, Australia, and other. Why did you decide to group the remaining participating countries into a single category ("other")? I think it would be interesting to know which other countries participated in the study.
The information is presented in tables and figures in an organized manner, making it easier for readers to understand.
On the other hand, I understand that the study is methodologically well-conceived, as is the discussion of the results and conclusions.
The authors are commended for including strengths, limitations, and future lines of research in the article.
Thank you very much.
Author Response
Dear Reviewer 3,
COMMENT 1: The topic is presented in the introduction in an orderly and organized manner, which aids understanding. You have used a considerable number of bibliographic references. However, a high number of them correspond to old publications, more than 5 years old. Therefore, you should search for more recent references and update them.
Response to reviewer:
Thank you for this suggestion. We have replaced older citations with citations for more recent work where more recent evidence was available and appropriate for the text it supported.
Change and location in manuscript:
The following are updated citations:
[9] Jardim TV, Mozaffarian D, Abrahams-Gessel S, Sy S, Lee Y, Liu J, Huang Y, Rehm C, Wilde P, Micha R, Gaziano TA,. Cardiometabolic disease costs associated with suboptimal diet in the United States: A cost analysis based on a microsimulation model. PLoS medicine. 2019;16(12), p.e1002981
[19] Nyberd, ST, Singh-Manoux, A, Pentti, J, Madsen, IE, Sabia, S, Alfedsson, L, Bjorner, JB, Borritz, M, Burr, H, Goldberg, M, Heikkilä, K. Association of healthy lifestyle with years lived without major chronic diseases. JAMA Internal Medicine. 2020;180(5): 760-768.
[20] MA, JK, McCracken, LA, Voss, C, Chan, FH, West, CR, Martin Ginis, KA,. Physical activity measurement in people with spinal cord injury: comparison of accelerometry and self-report (the Physical Activity Recall Assessment for People with Spinal Cord Injury).Disability and rehabilitation. 2020;42(2), 240-246.
[22] Olds, TS, Gommersall, SR, Olds, ST, Ridley, K., A source of systematic bias in self-reported physical activity: the cutpoint bias hypothesis. Journal of science and medicine in sport, 2019;22(8), 924-928.
COMMENT 2: Furthermore, it might be interesting to include more information in the introduction about the number of WAT users, both globally and in the countries participating in the study, both in the general population and in patients with chronic illnesses, as well as the personal characteristics of the users of these devices.
Response to reviewer:
Thank you for this suggestion. We have included more detail about how WAT users from the main countries included in this study, the general characteristics of users, and use in the general population as well as chronic disease.
Change and location in manuscript:
“Market research estimates approximately 20% of Australian adults [12] and 39% of US adults [13] own a WAT, with most users being between 25-44 years old, being tertiary educated or higher, and Apple Watches and Fitbits being the most used devices [14]. Modern day WATs have evolved from simple step counters to sophisticated devices that track numerous health metrics and integrate with smartphones to allow real time monitoring and personalized insights [15].
- Line 52-58, page 2
COMMENT 3: In Results, you group the data by country of the participant: United States, Australia, and other. Why did you decide to group the remaining participating countries into a single category ("other")? I think it would be interesting to know which other countries participated in the study.
Response to reviewer:
The ‘other’ countries category includes 31 different countries, with most of these only having just one participant and none having more than six participants. Due to the large number of other countries with very few participants, we grouped them as an ‘other’ category. We do agree with the reviewer that all of the countries represented is interesting information, and have therefore added a list of all countries and number of participants as a supplementary file.
Change and location in manuscript:
- Supplementary file 6 added
COMMENT 4: The information is presented in tables and figures in an organized manner, making it easier for readers to understand.
On the other hand, I understand that the study is methodologically well-conceived, as is the discussion of the results and conclusions.
The authors are commended for including strengths, limitations, and future lines of research in the article.
Response to reviewer:
Thank you for your feedback.
Reviewer 4 Report
Comments and Suggestions for Authors
Thank you to the authors for sharing the findings from this cross-sectional study on wearable activity trackers. The insights into users’ experiences with sharing tracker data in healthcare settings are informative and will likely be valuable to readers. However, I noted that a large proportion of participants were under 45 years of age and predominantly from the United States and Australia. These demographic characteristics may limit the generalisability of the findings to other age groups and countries. I recommend addressing this limitation more explicitly in the discussion.
Additionally, one important limitation of wearable activity trackers that was not sufficiently discussed is the reliability and validation of the devices and applications used. Activity data can vary based on factors such as the type of device, the specific app, and software versions. This variability may affect data accuracy. While the discussion covers privacy concerns, it does not address potential issues related to data accuracy or the possible harms of using inaccurate data for clinical decision-making. I suggest that the authors incorporate this perspective into the discussion section.
Overall, I found the study to be well-conducted and presented with a high level of quality.
Author Response
Dear Reviewer 4,
COMMENT 1: Thank you to the authors for sharing the findings from this cross-sectional study on wearable activity trackers. The insights into users’ experiences with sharing tracker data in healthcare settings are informative and will likely be valuable to readers. However, I noted that a large proportion of participants were under 45 years of age and predominantly from the United States and Australia. These demographic characteristics may limit the generalisability of the findings to other age groups and countries. I recommend addressing this limitation more explicitly in the discussion.
Response to reviewer:
We agree. We have now expanded the section of the discussion that acknowledged this limitation to further highlight it.
Change and location in manuscript:
“Though we recruited a global sample, some populations were over-represented. In particular, the sample in this study were mostly younger than 45 years, tertiary educated, and came from either Australia or the United States. The demographic characteristics of this sample means that the experiences and attitudes to data sharing identified may not be generalizable for the wider population or individuals from broader age, income or education groups.”
- Lines 320-326, page 9
COMMENT 2: Additionally, one important limitation of wearable activity trackers that was not sufficiently discussed is the reliability and validation of the devices and applications used. Activity data can vary based on factors such as the type of device, the specific app, and software versions. This variability may affect data accuracy. While the discussion covers privacy concerns, it does not address potential issues related to data accuracy or the possible harms of using inaccurate data for clinical decision-making. I suggest that the authors incorporate this perspective into the discussion section.
Response to reviewer:
Thank you for highlighting this. We have incorporated this suggestion in the discussion.
Change and location in manuscript:
“However, realizing this potential requires addressing user and provider barriers, and investment in technology, infrastructure, data security and clinician training. To use WAT data for clinical decision making, it is essential that data is sufficiently accurate. Currently, the validity and reliability of WAT data can vary across different devices and systems [41]. Identifying which devices meet the needs of healthcare delivery is critical to incorporate them into care, as is development of devices and systems that provide accurate data.”
- Line 341-347, page 10
COMMENT 3: Overall, I found the study to be well-conducted and presented with a high level of quality.
Response to reviewer:
Thank you for your feedback.
Reviewer 5 Report
Comments and Suggestions for Authors
Wearable activity trackers (WATs) are increasingly used by individuals to monitor physical activity, sleep, and other health behaviors. Integrating WAT data into clinical care may offer a cost-effective strategy to support health behavior change. However, little is known about users’ willingness to share their WAT data with healthcare providers.
AUTHORS aimed to explore attitudes and experiences of WAT users regarding the sharing of WAT data with healthcare providers, and examine how these vary according to user characteristics.
THEY proposed an online cross-sectional survey on adults who had used a WAT within the past three years.
The survey assessed user demographics, usage patterns, experiences of sharing data with healthcare providers, and willingness or concerns regarding data sharing. Multivariate logistic regression was used to examine associations between user characteristics and data-sharing experiences or attitudes. 447 participants completed the survey (60.0% female; 83.8% <45 years; 60.0% from the United States). Most (94%) participants expressed willingness to share WAT data with healthcare providers, 47% had discussed and 43% had shared WAT data in clinical settings. Privacy was the most commonly reported concern, cited by 10% of participants.
AUTHORS found that participants with chronic health conditions were more likely to have shared or discussed WAT data, but also more likely to report concerns.
Geographic differences were also observed, with Australian participants less likely to have shared or discussed their WAT data with providers, and US participants reporting fewer privacy concerns.
AUTHORS concluded that: the high willingness to share WAT data suggests untapped potential for integrating patient-owned WATs into clinical care.- Addressing privacy concerns and equipping healthcare professionals with the skills to use WAT data will be essential to fully realize this opportunity.
THEIR findings highlight the need for further development of secure WAT systems, clinician training, and expanded evidence on WATs’ clinical utility.
This is an interesting and well written manuscript.
I have the following comments with a pure academic spirit:
- In the introduction I would highlight how the approach to wearables has changed after the smartphone era.
- Also in the introduction I would avoid grouping the quotes and detail the individual contributions (see for example line 40 or line 63)
- The methods are described in detail and refer to the supplementary material for the survey, it would be nice to have a printout of the electronic survey in the article
- In electronic surveys there are many aspects to consider related to privacy management that are very complex. Please explain them.
- Insert a flow-chart that explains the flow of the design and avoid one-sentence paragraphs (see 2.3.1 and 2.3.2)
- I found the results and the discussion impeccable and very useful: even the paragraph that suggests future developments that I usually suggest in this case is already present
Author Response
Dear Reviewer 5,
COMMENT 1: This is an interesting and well written manuscript.
I have the following comments with a pure academic spirit:
Response to reviewer:
Thank you for providing feedback on our manuscript. Our responses are provided below.
COMMENT 2: In the introduction I would highlight how the approach to wearables has changed after the smartphone era.
Response to reviewer:
We have added detail in on how modern WATs have evolved to integrate with smartphones as suggested.
Change and location in manuscript:
“Modern day WATs have evolved from simple step counters to sophisticated devices that track numerous health metrics and integrate with smartphones to allow real time monitoring and personalized insights [15].”
- Line 55-57, page 2
COMMENT 3: Also in the introduction I would avoid grouping the quotes and detail the individual contributions (see for example line 40 or line 63)
Response to reviewer:
Thank you for this suggestion. We have reduced the grouping of citations throughout the introduction where applicable as recommended.
Change and location in manuscript:
“Unhealthy lifestyle behaviors such as physical inactivity and sedentary behavior [1], poor sleep quality [2], and poor diet quality [3] contribute substantially to global morbidity and mortality.”
- Line 38-40 page 1
“Poor diet quality [5] and insufficient sleep [6, 7] are also widespread and linked to similar health risks.”
- Line 43-44, page 2
“Previous research indicates that individuals with higher education, higher income, and diagnosed chronic conditions may be more open to sharing their health data [27], while underrepresented minorities and those with lower educational attainment or poorer self-rated health may be more hesitant [28].”
- Lines 74-77, page 2
“Existing studies have often focused on data sharing for research rather than clinical use [30], have been limited to single-country samples [31], or were conducted when WATs were less common [32].”
- Lines 80-83, page 2
COMMENT 4: The methods are described in detail and refer to the supplementary material for the survey, it would be nice to have a printout of the electronic survey in the article.
Response to reviewer:
We provided the survey as a supplementary file and not in the manuscript due to its length (9 pages). The key questions from the survey that this manuscript presents results for are included in Figure 1.
COMMENT 5: In electronic surveys there are many aspects to consider related to privacy management that are very complex. Please explain them.
Response to reviewer:
As suggested, we have added detail in the methods regarding privacy considerations for electronic surveys, and how this was managed in this study.
Change and location in manuscript:
“As online surveys often collect metadata such as IP addresses and tracking links, true anonymity is difficult to guarantee. Therefore, we only collected data that was necessary for the purpose of the survey, we did not collect sensitive information, and access to study data was restricted only to the researchers.”
- Line 99-102, page 3
COMMENT 6: Insert a flow-chart that explains the flow of the design and avoid one-sentence paragraphs (see 2.3.1 and 2.3.2)
Response to reviewer:
(a) We don’t believe that a flow chart is appropriate for illustrating the flow of the study design, as this is a cross-sectional study with only one stage. The wording in the Materials and Methods section has been edited to explain the study design more clearly.
(b) We have edited the materials and methods section to avoid using one-sentence paragraphs.
Change and location in manuscript:
(a) “The data presented in this study are from a sub-set of questions in a larger cross-sectional survey that explored WAT users’ experiences related to device usability and perceived behavior change [14]. Due to the size of the larger survey and the different topic of enquiry, the subset of data presented in this study have been reported separately.”
- Line 92-96, page 3
(b) “Participants’ demographic characteristics were collected, including gender, age, country of origin, self-rated health status and the presence of chronic health conditions. Participants were asked to report if they were a current or former user of a WAT, the brand of device they used, how long and how frequently they used the WAT, and if they perceived any change in physical activity levels since using a WAT.”
- Line 124-128, page 3
“Participant and WAT characteristics were summarized using descriptive statistics (n and %). Responses to the four WAT data-sharing survey items were analyzed using descriptive statistics, and free-text responses were categorized thematically. Subgroup analyses were conducted using multivariate logistic regression to examine associations between WAT data-sharing attitudes and participant characteristics, including gender, age, country of residence, education level, self-rated health, and presence of chronic conditions. Given the number of comparisons, unadjusted p-values are reported, and results are interpreted cautiously due to the increased risk of Type I error associated with multiple testing. Events-per-variable (EPV) ratios were reviewed for each of the four models. All statistical analyses were performed in Stata 18 (StataCorp, College Station, TX, USA), with statistical significance set at p < 0.05. Figures were generated using MATLAB version 24.1 (MathWorks, Natick, MA, USA).”
- 145-156, page 4
COMMENT 7: I found the results and the discussion impeccable and very useful: even the paragraph that suggests future developments that I usually suggest in this case is already present.
Response to reviewer:
Thank you for your feedback.
Round 2
Reviewer 2 Report
Comments and Suggestions for Authors
Thank you for addressing the comments
Reviewer 3 Report
Comments and Suggestions for Authors
Dear authors,
I appreciate the work you have done in responding to my suggestions to improve the methodological quality of the manuscript, as well as for clarifying any questions or comments I made.
I congratulate you on your work.